# Cement-Based Composites Containing Oxidized Graphene Nanoplatelets: Effects on the Mechanical and Electrical Properties

**DOI:** 10.3390/nano13050901

**Published:** 2023-02-27

**Authors:** Luca Lavagna, Andrea Santagati, Mattia Bartoli, Daniel Suarez-Riera, Matteo Pavese

**Affiliations:** 1Department of Applied Science and Technology, Politecnico di Torino, C.so Duca degli Abruzzi 24, 10129 Torino, Italy; 2Consorzio Interuniversitario Nazionale per la Scienza e Tecnologia dei Materiali (INSTM), Via G. Giusti 9, 50121 Florence, Italy; 3Center for Sustainable Future, Italian Institute of Technology, Via Livorno 60, 10144 Turin, Italy; 4Department of Structural, Geotechnical and Building Engineering, Politecnico di Torino, C.so Duca degli Abruzzi 24, 10129 Torino, Italy

**Keywords:** graphene nanoplatelets, cement-based composites, oxidation, mechanical properties, electrical properties

## Abstract

Graphene nanoplatelets can improve the electrical and mechanical properties of cement matrix composites. The dispersion and interaction of graphene in the cement matrix appears to be difficult due to its hydrophobic nature. By introducing polar groups, graphene oxidation improves the level of dispersion and interaction with the cement. In this work, graphene oxidation using sulfonitric acid for 10, 20, 40, and 60 min was studied. Thermogravimetric Analysis (TGA) and Raman spectroscopy were employed to analyze the graphene before and after the oxidation. The mechanical properties of the final composites showed an improvement of 52% in the flexural strength, 4% in the fracture energy, and 8% in the compressive strength in the case of 60 min of oxidation. In addition, the samples showed a reduction of at least one order of magnitude in electrical resistivity when compared with pure cement.

## 1. Introduction

Graphene is a two-dimensional (2D) material of only one atomic layer thickness, composed of carbon atoms with sp^2^ hybrid orbitals and presents a honeycomb-like lattice structure. Since its discovery in 2004, graphene has received considerable scientific interest for its unique behavior and excellent thermal, electrical, optical, and other properties [1,2,3,4]. “Graphene is the strongest material ever tested”; with this sentence Lee et al. [5] declared the extraordinary mechanical properties of this material with an intrinsic tensile strength of 130 GPa and a Young’s modulus of 1 TPa. The use of graphene in composite materials can improve their electrical [6], thermal [7], and mechanical properties [8,9]; however, graphene applications are currently mainly focused on the new generations of electronic devices [10,11]. Yet, in the field of cements and concretes the use of graphene has long been known [12,13,14]. Numerous articles report important improvements in both the mechanical strength and electrical conductivity of cement or concrete [15,16,17]. However, one of the main problems related to the use of such nanomaterials is their dispersion in the matrix [18]. Generally, graphene is dispersed in the water used for cement preparation, but its carbonaceous and extremely hydrophobic nature makes it difficult to achieve a stable and homogeneous dispersion [19,20]. In addition, the interaction with the matrix is found to be low due to the chemical incompatibility [21]. Several studies have shown that the use of graphene oxide (GO) instead of graphene can be a possible solution to overcome these problems [22]. Not only GO is more dispersible in water, but within the cement paste, GO seems to have a much better interaction with the matrix [23]. However, as some works have shown, in an extremely basic environment such as cement, GO agglomerates, leading to a deterioration of the electrical and mechanical performance of the cement matrix composites [24,25]. One possible solution is the functionalization of graphene to improve its dispersion in water and interaction with the cement paste [26,27]. It is known that carboxyl groups improve the interaction and dispersion in the cement matrix [28,29], but the best oxidation time to achieve an improved mechanical and electrical performance is currently unclear. The goal of this paper is to determine the optimum oxidation time to obtain an improvement from both the mechanical and the electrical points of view.

It is currently difficult to identify in the literature which GNPs’ oxidation time achieves the highest improvement of both the mechanical and electrical properties of the cement matrix composites.

## 2. Materials and Methods

Grade 4 commercial graphene nanoplatelets (GNPs), purchased from Cheaptubes, Grafton, MA, USA, (average X and Y dimension < 2 µm, average thickness 8–15 nm, purity 97%) were chemically oxidized by reaction with a 3:1 mixture of sulfuric acid 98% (H_2_SO_4_) and nitric acid 65% (HNO_3_), all from Merck, St. Louis, MO, USA. To stop the oxidation reaction at the required time, sodium hydroxide (NaOH, Merck, 97%) was used to neutralize the pH. The cement powder used in this study was an ordinary Portland cement 52.5 R Ultracem purchased from Italcementi S.p.A., Bergamo, Italy, Table 1 shows the composition of the cement used.

A Mettler Toledo 1600, Columbus, OH, USA was employed for the thermogravimetric analysis. The samples were heated with a constant heating ramp of 10 °C/min from 25 °C to 1000 °C. The air was supplied at a constant flow rate (50 mL/min).

The Raman spectra were recorded using a Renishaw inVia (H43662 model, Gloucestershire, UK) equipped with a green laser line (514 nm) with a 50× objective in the range from 3500 cm^−1^ to 250 cm^−1^.

The experimental setup for the electrical resistivity was composed of copper cylinders (5 cm diameter and 5 cm length), thin copper foil, and a hydraulic press (Specac Atlas Manual Hydraulic Press 15T). The samples were positioned between the aligned copper cylinders in contact with the copper foil. The methodology used for the measurement was reported in literature [30]. To avoid electrical interference, insulating sheets were placed between the conductive cylinders and the load. An Agilent 34401A multimeter was used to measure the resistance of the composite samples. 

A single-column Zwick-Line z050 with a load cell of 1 kN was used for the flexural and compressive tests. The flexural test was measured with a three-point bending test in crack mouth opening displacement (CMOD); the samples were prepared following the standard JCI-001-2003, with a notch in the middle of the sample of 5 mm in depth and a width of 2 mm. An extensometer was placed at the two sides of the notch, and the CMOD was controlled at a fixed rate of 0.005 mm/min with a span of 65 mm. This setup allows measurement of not only the flexural strength but also the fracture energy as described in the literature [31].

Compressive tests were performed using the same machine with a 50 kN load cell. The test was conducted on one of the two halves produced by the flexural test. The test was conducted with a speed of 600 N/s with a preload of 20 N. Both the flexural test and the compressive test results were an average of at least four specimens. The morphology of the cement and cement composites containing GNPs were observed with a scanning electron microscope (SEM), FEG ASSING SUPRA 25.

A caliper was used to measure the width, length, and height, and the samples were weighted on a RADWAG PS 510/C/1 analytical balance.

## 3. Experimental Section

In the oxidation experiments, 0.5 g of GNPs was dissolved in 60 mL of a solution of sulfonitric acid (3 H_2_SO_4_: 1 HNO_3_). This reaction led to the formation of carboxyl groups on the surface of the GNPs, as shown in the literature [28,32]. The acidic solution containing the GNPs was placed in an ultrasonic bath (SONICA 2400MH series) for different oxidation times (10, 20, 40, and 60 min) to ensure a continual agitation of the suspension during the reaction. To stop the oxidation process, a 1 M NaOH solution was prepared to neutralize the acid. The solution was then filtered on a G4 sintered glass filter and washed several times with deionized water. The GNPs were then dried in an oven at 80 °C overnight. Both pristine and oxidized GNPs were used to prepare the cement-based composites containing 0.1% bwoc (by weight of cement) of GNPs. The value was chosen as low as possible in order to limit the costs and dispersion issues and to remain under the typical percolation threshold for this type of carbon nanomaterial, as suggested in literature for applications that involve sensing [24,33,34,35,36,37]. Moreover, samples of pure cement were used for comparison. The mix design of the prepared specimens is reported in Table 2.

The procedure to prepare the cement composites consisted in the redispersion of the GNPs in water for 15 min with an ultrasonic tip (VibraCell™) at 100 W power. The dispersion of the GNPs in water was then mechanically stirred for several min, while cement powder was slowly added. The cement paste, with a water-to-cement (w/c) ratio of 0.5, containing 0.1% bwoc of GNPs, was then poured into suitable molds and cured for 24 h at 85 °C in a controlled environment, at 100% relative humidity (Figure 1).

This high curing temperature, used for instance in cement samples preparation for oil and gas applications, enables completion of the hydration reaction of cement in only 24 h, so that more tests can be carried out in a shorter time. Prismatic molds of size 20 × 20 × 80 mm were used for the cement composites. The results of the tests are shown in Table 3.

## 4. Results and Discussion

The TGA curves shown in Figure 2A display different thermal stability in flowing air as a function of the oxidation time of the samples. In fact, as can be seen, oxidation for 20 min (red line) shifted the onset of the curve at a slightly lower temperature than the pristine GNPs (black line). The onset of the curve for 60 min of sulfonitric treatment started at a much lower temperature, about 100 °C less than the pristine GNPs. This was due to the presence of the carboxyl functional groups that tended to lower the thermal resistance to degradation [32].

Indeed, the acidic treatment tended to damage the graphitic lattice structure of the graphene. This effect was confirmed by Raman analysis (Appendix A). The value of the ratio between the I_D_ and I_G_ bands (I_D_/I_G_) (Figure 2B), which indicated the extension of the graphitic defects [38], showed a decrease after oxidation for 10 min. This was probably due to the fact that short oxidation times did not affect the graphitic structure but rather cleaned off the amorphous carbon present on the surface [39,40]. Thereafter, the value tended to increase, as expected due to the creation of defects on the graphitic layer. The I_D_/I_G_ ratio reached its maximum after 40–60 min of oxidation.

As shown in Table 2 and Figure 3, the best results from the mechanical and electrical point of view were obtained with the GNPs oxidized for 60 min. This treatment led to a 52% improvement in the flexural strength, a 4% improvement in the fracture energy, an 8% improvement in the compressive strength, and a reduction in the electrical resistivity of 94% compared to the plain cement (Figure 2).

The use of pristine GNPs not only did not increase the mechanical properties of the cement, but a reduction was instead observed (Figure 3A). This result was due to the fact that the dispersion in water and in the cement matrix was not optimal, notwithstanding the sonication procedure; so, agglomerates were present, acting as defects (Appendix A). Moreover, the interaction between the reinforcement and the matrix was weak due to the extremely hydrophobic nature of carbon. Oxidation for 10 min had a further negative effect, with a decrease in the compressive strength and fracture energy and an increase in the resistivity. This can be explained by the removal of the amorphous carbon impurities after the first min of oxidation. The amorphous carbon present on the surface of the pristine GNPs seemed to have a positive effect, probably increasing the anchorage with the cement matrix due to the presence of some oxide functionalities. After 10 min of oxidation, this layer was removed, and the GNPs presented an improved level of graphitization, as confirmed by the Raman data (Figure 2B).

Further oxidative functionalization improved the interaction of the GNPs with the cement matrix; in fact, by increasing the oxidation time, the mechanical properties, in particular the flexural strength, tended to increase again. After oxidation for 60 min, the samples showed significantly better flexural strength than the pure cement, and the compressive strength and fracture energy increased, even if only slightly. This increase was probably due to the presence of an adequate amount of oxygenated groups on the surface of the GNPs, which resulted in a better dispersion (Appendix A) and interaction with the cement matrix [24]; agglomeration was not observed even in the case of strongly basic conditions (Appendix A). 

Indeed, from the SEM analysis, a morphological change in the microstructure of the cement, in particular the C-S-H gel (calcium silicate hydrates), was observed, that changed from a more compact structure (Figure 4A,B) to a more filamentous structure (Figure 4C,D red arrow). This transformation was suggested by Zheng et al. [41], who assumed that graphene acted as the nucleation point of the C-S-H gel during hydration. This fact was also confirmed by the increase in the samples’ density as a function of the oxidation time, leading to a decrease in porosity of about 8 percent compared to that of the cement sample, confirming that, at a higher oxidation time, the interaction with the matrix improved (Table 2).

In contrast, the samples sonicated for 20 and 40 min did not show a clear improvement in flexural strength, and both the compressive strength and fracture energy were lower than the pure cement; the number of functional groups present on the surface of the reinforcement was probably not sufficient to compensate for the presence of the defects, and a longer treatment was needed to achieve a good dispersion and interaction between the GNPs and the cement.

In the case of the compression strength and fracture energy, it was particularly evident how graphene’s presence can even worsen the mechanical strength. As stated above, short acidic treatments remove impurities from the surface, thus leading to less interaction with the cement paste. Furthermore, although the carbon reinforcements are typically effective in hindering crack propagation, the addition of these materials can create defects, which adversely affect the compressive strength of the composites, as shown by Lining et al. [42]. Even worse is the effect of not sufficiently oxidized GNPs on the fracture energy. Since graphene has a platelet form, the crack bridging effect is not very good, as in the case of carbon fibers or carbon nanotubes. Moreover, the lateral size of the GNPs used in this study was very low, and the fracture toughness (and consequently the fracture energy) was positively affected by the lateral size of the graphene nanoplatelets [43].

The use of graphene nanoplatelets within the cement matrix always had a positive effect on the electrical resistance of the cement, which decreased significantly even in the case of the pristine GNPs. While in the case of fibers and nanotubes [34,44], the oxidation seems to have a profound effect in reducing the electrical resistivity of the composites, in this case, the oxidation of the GNPs did not have a significant influence. However, the reduction in the electrical resistivity observed in the composites with respect to the pure cement could provide greater sensitivity for structural monitoring.

## 5. Conclusions

In this work, the surface oxidation of GNPs was shown to have a beneficial effect on improving the electrical and mechanical properties of the cement matrix composites. The oxidation did not appear to have excessively damaged the lattice structure of the GNPs but most probably ensured the presence of carboxyl groups on the surface. These allowed a better dispersion of the GNPs in water and thus in the final composite, as well as a better interaction between the matrix and the GNPs. The oxidation treatment must however be long enough to guarantee the presence of a significant number of functionalities on the GNPs’ reinforcement; otherwise, there is a risk of lowering the compression strength and fracture energy with respect to the pure cement. In this paper, the maximum beneficial properties resulted from 60 min of sulfonitric acid treatment.

The use of graphene or other carbon-based reinforcement inside cement-based materials is increasingly interesting for possible applications in the field of the continuous monitoring of structures, where a reduction in the electrical resistance of cement is essential to increase the sensitivity of the measurements. These carbon–cement composites that contain a very small amount of carbon and thus have an acceptable cost will probably open the door to the widespread monitoring of structures, increasing their safety and durability.

## Figures and Tables

**Figure 1 nanomaterials-13-00901-f001:**
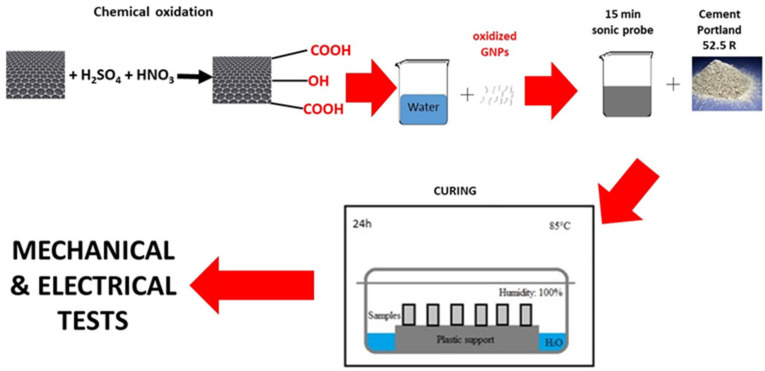
Preparation route of the samples.

**Figure 2 nanomaterials-13-00901-f002:**
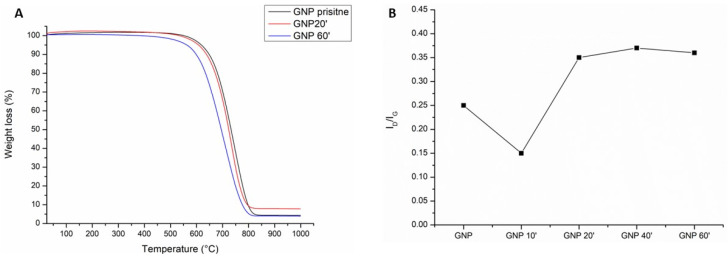
The thermal gravimetric analysis (**A**) of the pristine GNPs (black line) and after oxidation with sulfonitric acid for 20 min (red line) and 60 min (blue line); the Raman I_D_/I_G_ ratio vs the oxidation time (**B**).

**Figure 3 nanomaterials-13-00901-f003:**
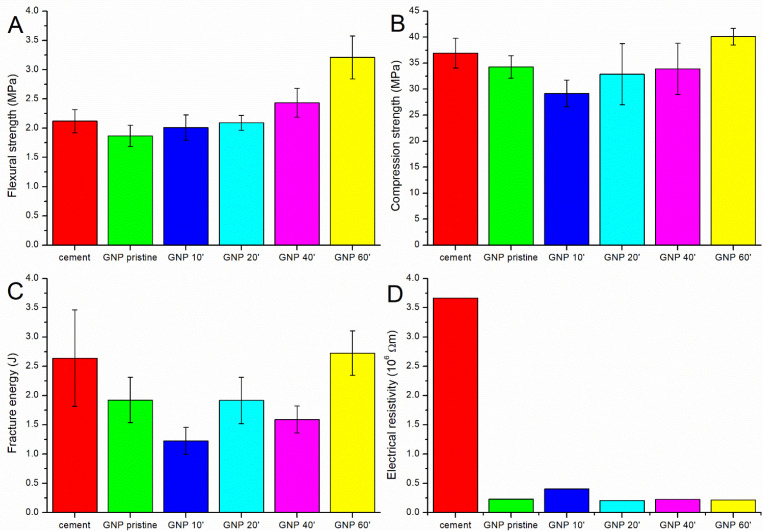
The (**A**) flexural strength, (**B**) compression strength, (**C**) fracture energy, and (**D**) electrical resistivity of the pure cement and of the cement containing pristine and oxidized GNPs.

**Figure 4 nanomaterials-13-00901-f004:**
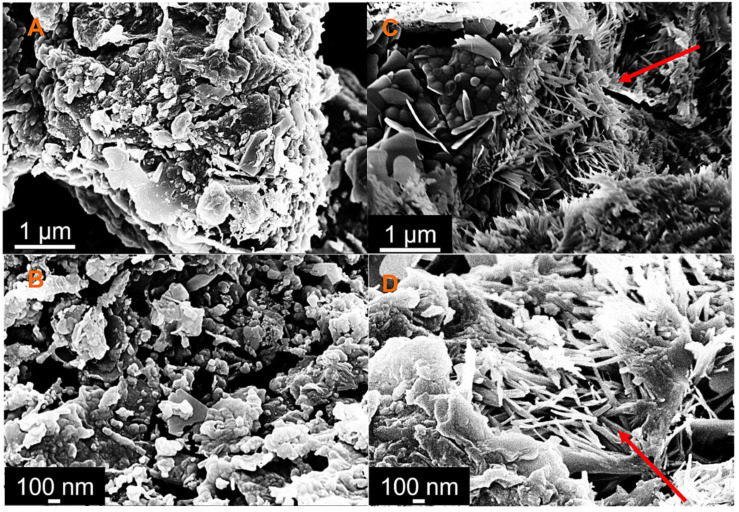
The SEM images of the cement (**A**,**B**), and the cement with GNPs oxidized for 60 min (**C**,**D**).

**Table 1 nanomaterials-13-00901-t001:** The composition and properties of the ordinary Portland cement 52.5 R.

Oxide	(wt.%)	Phase	(wt.%)
SiO_2_	20.0	C3S	49.1
CaO	63.2	C2S	19.7
Al_2_O_3_	4.1	C3A	7.9
Fe_2_O_3_	1.9	C4AF	5.2
MgO	4.2		
SO_3_	3.4		
Na_2_O	0.003		
K_2_O	0.001		
Loss on ignition	0.8		

**Table 2 nanomaterials-13-00901-t002:** Mix design of the prepared cement-based composites.

Sample Name	Water (g)	Cement (g)	GNP (g)
Cement	95.25	190.5	/
GNP	95.25	190.5	0.19
GNP 10’	95.25	190.5	0.19
GNP 20’	95.25	190.5	0.19
GNP 40’	95.25	190.5	0.19
GNP 60’	95.25	190.5	0.19

**Table 3 nanomaterials-13-00901-t003:** In the columns are reported, from left to right, the code of the sample (treatment/time), the time of the acid treatment, and the mechanical and electrical performance of the cement composites (pure cement or containing 0.1% bwoc of the pristine and oxidized GNPs) resulting from each test.

Sample Name	Time (min)	Density (g/cm^3^)	Flexural Strength (MPa)	Fracture Energy (J)	Compression Strength (MPa)	Electrical Resistivity (10^6^ Ω m)
Cement	/	1.63	2.1 ± 0.2	2.6 ± 0.8	36.9 ± 2.9	3.6
GNP	/	1.62	2.2 ± 0.2	1.9 ± 0.4	34.3 ± 2.2	0.2
GNP 10’	10	1.70	1.8 ± 0.2	1.2 ± 0.2	29.2 ± 2.6	0.4
GNP 20’	20	1.76	2.0 ± 0.1	1.9 ± 0.4	32.9 ± 5.9	0.2
GNP 40’	40	1.73	2.5 ± 0.2	1.6 ± 0.2	33.9 ± 4.9	0.2
GNP 60’	60	1.86	3.3 ± 0.3	2.7 ± 0.3	40.1 ± 2.5	0.2

## Data Availability

The data presented in this study are available on request from the corresponding author.

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
