# Peer review of "Cement-Based Composites Containing Oxidized Graphene Nanoplatelets: Effects on the Mechanical and Electrical Properties"

_nanomaterials, 2023, doi:10.3390/nano13050901_

Round 1
Reviewer 1 Report
Comments:
This is the study of the mechanical and electrical properties of cement-based composites with oxidized graphene nano-platelets.
I do not have any major concerns about significance and technical quality of this research and would recommend its publication after minor revision – see my comments below.
In Figure 2(B), authors show the Raman ratio ID/IG as a function of oxidation time up to 60 minutes.I wonder whether the ID/IG reaches the maximum or not at 60 minutes. Why don’t you observe the Raman ratio at the oxidation time over 60 minutes?
In page 5 at line#173, the term C-S-H should be written by a full name.
In Figure 4(B), authors mention that the photo shows a more filamentous structure.
It is better that the sites of the filamentous structure are indicated with an arrow in the photo.
Mistype setting: In page 2 line # 82, CMOD, line#83, 2mm. and line# 85, non.
Reviewer 2 Report
This manuscript presents an experimental study on the mechanical and electrical properties of CNPs cement composites. Overall, the writing is good and this study provides some useful information for the research community. Some small comments are given to improve the quality of this communication.
# As graphene-based cement composites have been widely investigated about ten years ago, the innovation of this research must be highlighted, and the difference between this research from previous contributions should be pointed out in detail.
# Please present the detailed mix proportions of the samples in tables.
# As CNPs need a long time for excellent dispersion using an ultrasonic bath, please indicate how can this material be used in large-scale practical applications.
# The direct/indirect observation of the dispersion of CNPs will help highly improve the quality of this manuscript.
# It would be nice if more analyses on the porosity and reaction products of the paste can be given.
Reviewer 3 Report
This work discusses the use of graphene nanoplatelets to improve the electrical and mechanical properties of cement matrix composites. Although the work is well written, I think there some experiments need to be done and some comments to be taken into consideration.
1- The introduction is written well, however, I couldn’t see the motivation of the work
2- Could you please explain the novelty of the work? What is your novelty.
3- In the lines of 55 and 56 the authors stated that “The goal of this article is to find which oxidation conditions are the best in order to obtain an improvement from both the mechanical and electrical point of view. However, the authors study the oxidation time only and they didn’t change any other parameter.
4- The study needs to be done at different weight ratio of GNP (0.4, 0.6, 0.8 and 1 wt.%, not only one wt.% in order to see an effect as I think 0.1 wt.% is not enough to make a conclusion.
5- Table 2 has a column talking about density, but there is no mention in the text to explain about it.
6- Figure 2 B shows “according to your explanation” that 40 and 60 minutes have the same degree of oxidation. However, table 2 shows that GNP 40’ and GNP 60’ have different results. Is there any explanation for that?
7- Is there any explanation for the electrical resistivity improvement
8- I believe you better do elemental analysis and corelate with oxygen content not oxidation
9- Could you provide Raman plots
10- It seems that plain cement has better properties than cement with other GNP, only the GNP 60’ showed improvement. Could you please explain?
11- I think the table should report about concrete not cement!
12- Figure 4 the SEM images quality is bad, could you provide clear images (not charging)
Round 2
Reviewer 3 Report
You better make it a comprehensive paper; I don’t see that you considered the comments seriously. Anyway, I just want to let you know that I have spent my valuable time reading your manuscript to improve it and make it a better manuscript (scientifically), and it is a work that as a reviewer I do it for free if you know that! so believe me I am not here to challenge you or make it hard for you. At the end it is your choice.
